# Distinguishing Curable from Progressive Dementias for Defining Cancer Care Options

**DOI:** 10.3390/cancers15041055

**Published:** 2023-02-07

**Authors:** Catherine H. Schein

**Affiliations:** Department of Biochemistry and Molecular Biology, Institute for Human Infections and Immunity, University of Texas Medical Branch at Galveston, Galveston, TX 77555, USA; chschein@utmb.edu

**Keywords:** conditions causing altered mental state, dementia, delirium, depression (3Ds), cancer care team, polypharmacy, de-prescribing, PIMs (potentially inappropriate medications), neurological syndromes, Alzheimer & Parkinson diseases, cancer genetic profiling, non-surgical intervention in frail patients, immunotherapy in CRC, microsatellite instability (MSI/DNA mismatch repair (dMMR), autoimmune encephalitis

## Abstract

**Simple Summary:**

The median age of diagnosis of dementia, as well as for many cancers, is above 60 years. Dementia should not exclude even frail, elderly patients from cancer treatment. Before beginning any treatment, the effects of the patient’s diet, medications and caregiver situation on neurological symptoms should be determined. Malnutrition, dehydration, alcohol consumption, and even loneliness can all accentuate or cause the “3Ds” of dementia, delirium and depression. Common drugs, especially if taken together, can cause cognitive difficulties resembling neurodegenerative diseases such as Alzheimer or Parkinson. These syndromes may be reversed by diet, social and caregiver changes, and stopping or lowering the dose of potentially inappropriate medications (PIMs). Brain scans and genetic analysis can be used to guide immunotherapies and tumor-antigen directed treatments. Discussion among the patient, caregivers and medical team is needed to assess cognition state and ability to accept diagnostic and therapeutic procedures. While surgery may be too dangerous for frail patients, radiation and many oral or infusion therapies, some discussed in this review, may be better tolerated.

**Abstract:**

The likelihood of a diagnosis of dementia increases with a person’s age, as is also the case for many cancers, including melanoma and multiple myeloma, where the median age of diagnosis is above 60 years. However, patients diagnosed with dementia are less likely to be offered invasive curative therapies for cancer. Together with analysis of diet and medication history, advanced imaging methods and genetic profiling can now indicate more about syndromes causing the neurological symptoms. Cachexia, malnutrition, dehydration, alcohol consumption, and even loneliness can all accentuate or cause the “3Ds” of dementia, delirium and depression. Many common drugs, especially in the context of polypharmacy, can cause cognitive difficulties resembling neurodegenerative disease. These syndromes may be reversed by diet, social and caregiver changes, and stopping potentially inappropriate medications (PIMs). More insidious are immune reactions to many different autoantigens, some of which are related to cancers and tumors. These can induce movement and cognitive difficulties that mimic Alzheimer’s and Parkinson’s diseases and other ataxias associated with aging. Paraneoplastic neurological syndromes may be reversed by directed immunotherapies if detected in their early stages but are best treated by removal of the causative tumor. A full genetic workup should be done for all individuals as soon as possible after diagnosis, to guide less invasive treatments suitable for frail individuals. While surgical interventions may be contraindicated, genetic profile guided immunotherapies, oral treatments, and radiation may be equally curative in a significant number of cancers.

## 1. Introduction

The diagnosis of cancers increases with age, to over 1000/100,000 (1%) in people over 60 [1]. In the UK, a third of new cancers are diagnosed in those over 75 years of age [2]. The median age of diagnosis for many cancers is also over 60 (Table 1). At the same time, the incidence of mild cognitive impairment (MCI), that typically does not interfere with daily life, in the US increases steadily over the age of 58, peaking in those > 80 years of age. Although memory loss and behavioral changes, dementia, delirium, and depression (the 3Ds, Figure 1) may be psychiatrically attributed to Alzheimer’s (AD) and Parkinson’s (PD) diseases [3], there are many other syndromes that present with similar symptoms [4].

The recent advent of possible treatments, beginning with the MCI phase of AD, makes it imperative to define potentially treatable dementias, some of which may mimic AD or PD [5]. When faced with a patient suffering from cancer and the 3Ds, it is thus wise to consider their diet, social situation, medications, and autoimmune conditions before deciding on treatment. A rapid onset of neurological symptoms, as opposed to a years-long progressive loss of memory, may distinguish some of the reversible syndromes [6].

## 2. Problems Specifically Related to Cancer Patients with Dementia

Cancer diagnosis in patients with dementia is likely to occur at a later stage in the disease compared to those without, which may account for its higher lethality in this population. Those with dementia may be unable to effectively register pain [7]. After diagnosis, those with cancer are less likely to be offered invasive treatment such as surgical removal of a tumor, even if this is more likely to lead to a cure (in some cases, to both the cancer and dementia, see below). One of the reasons for this is that major surgery may shorten life expectancy. In a study of about 1200 major surgeries in patients over 65, 1-year mortality was around 6% for those not frail or exhibiting signs of dementia. However, these numbers increased drastically, to as high as 30% in those with probable dementia [8]. While avoiding major surgery makes sense with frail patients, those classified as having AD are also less likely even to receive palliative care [9].

Treatment of any cancer patient can require visiting multiple doctors and waiting for long periods of time to be seen. This can be an undue burden for those living with memory and movement disorders. It is thus essential to determine at a very early stage whether the patient will be able to understand enough to accept diagnostic testing and participate in eventual therapies. The caregivers and medical staff should be able to present their points of view on this matter freely. Even MCI can cloud a person’s ability to remember to take an oral medication or visit a series of doctors’ offices or treatment facilities on schedule. In addition to those specifically related to their cancer, patients may also need to remember medications for other syndromes, such as diabetes, heart disease, high blood pressure, and kidney function.

Assuming the help of family members or other caregivers in all aspects has been agreed to, they should keep a large print calendar and allocate the daily required medications in a way the patient can understand. Caregivers should receive copies of all documents related to the prescribed treatments and be included in doctors’ visits. They should also be allowed to attend when the patient is undergoing any treatment or diagnostic procedure [2]. However, one should also help the helper, by combining the visits as much as possible, thus reducing their number. Also, some member of the patient’s care team should periodically review all the medications, paying particular attention to those that could worsen cognitive difficulties. The helper should also answer:

## 3. What Is the Patient Eating and Drinking?

Non-pharmaceutical solutions may ameliorate delusional states and dementia related to environmental and caregiver problems [10]. Even simple dehydration can adversely affect cognition [11], resulting in the 3Ds. Malnutrition, often accentuated by loneliness or lack of oversight, can cause a range of neurological syndromes [12]. The excess consumption of alcohol can lead to vitamin deficiencies and dehydration, which can resemble other dementias and ataxias [13]. Depression, or not finding food tasty when moved to a care facility, can also lead to malnutrition and vitamin deficiencies [14].

## 4. What Medications and Supplements Are the Patient Taking?

One study found that 79% of older Americans living at home were taking medications that could affect cognition [15]. Often, the 3Ds can be related to interactions between the medications and supplements a patient is taking. Many older patients are taking drugs to treat chronic conditions, diabetes, high blood pressure, heart or kidney disease, gout or rheumatoid arthritis, psoriasis, allergies, or to prevent blood clots. On top of prescribed medications, they may also be taking over the counter (OTC) vitamins or other nutrition aids, NSAIDs, benzodiazepines to help them sleep, a proton pump inhibitor for acid reflux, or a seemingly innocuous senna preparation against constipation (possibly brought on by lack of movement or an opioid prescription for pain). They may also be taking something recommended in their native country to enhance “potency”. The number of drugs and supplements can rapidly add up: many studies have shown “polypharmacy”, defined as taking 5–10 drugs simultaneously, is extremely common in the elderly population. Many of these drugs, alone or in combination, can induce delirium [16].

## 5. PIMs and Polypharmacy

The “Beers Criteria”, named after the geriatrician who focused on the problem of polypharmacy in the elderly, originated with the realization that, when combined, many of the drugs were potentially inappropriate medications (PIMs). The PIM list [17] is too long to be included here. The PIM designation does not mean that the drugs should never be prescribed, but that care should be taken, especially in older persons and in a polypharmacy setting. The more dangerous drugs, due to their lesser advertised potential side effects, include diuretics to control blood pressure and first-generation antihistamines. Diuretics can lead to dehydration and resultant cognitive difficulties [18]. Common side effects of first-generation antihistamines include sleepiness (which can impair driving) and behavior changes. The list of drugs with potential interactions with promethazine are particularly troubling, including sedatives, muscle relaxers, antidepressants, atropine, blood thinners, MAO inhibitors, and medications to treat PD, ulcers, or IBS (taken from the longer list at https://www.rxlist.com/phenergan-side-effects-drug-center.htm#overview) Accessed on 3 February 2023. Promethazine is still available as an OTC generic in the US, although a prescription is required in many countries [19]. The adverse effects associated with later generation antihistamines are considerably lower, but they should be considered when combined with other drugs or supplements.

## 6. Is It Really Alzheimer’s Disease (AD)?

In previous times, AD could only be diagnosed post-mortem, based on staining for plaques in sectioned brain tissue. We are now armed with better tools for visualizing the brains of patients, which may with time yield better clues to disease progression and treatment options for co-morbidities [20]. Brain imaging is non-invasive, but the ability of the patient to be constrained and stay immobile for some time, as well as the ability to accept the intervention, must be discussed with the patient and caregivers. Neuroimaging techniques have recently been developed to clearly distinguish different forms of AD and PD [21]. Improved specific dyes for PET imaging can clearly indicate Aβ plaques and tangles in brain scans, a technique that has greatly clarified patient populations for clinical tests of new treatments for these diseases. MRI or PET scans of those with a clinical neurological AD diagnosis may reveal a much more complicated situation. Cerebrovascular disease, and hippocampal sclerosis with aging [22], may be more common causes of dementia in those over 75 years of age [23]. AD dementia may be combined with many other causes, such as small (or micro-) vessel ischemic disease (SVID and others, see Table 2 for a partial list) [24].

Cerebrospinal fluid markers may also aid in estimating disease progression [25]. Many of the related syndromes of Table 2 can have slower prognoses or promise possible improvement, which may be important when financial considerations are so based on remaining years of life estimates. Indeed, white matter hyperintensity on MRI, the progression of which usually accompanies cognitive decline, was stable or reversed (“white matter regression”) in almost half of AD patients followed in one study. Those with white matter regression also showed improved cognition [26].

These distinctions become increasingly important for selecting appropriate treatments beyond diet and exercise. Until recently, there was a general lack of curative treatments beyond the symptomatic or anti-inflammatory [27] for neurodegeneration in AD and PD. However, there is a continued push to test new therapies [28,29,30,31,32], and repurposed older ones [33,34,35]. For example, gemfibrozil, a peroxisome proliferator activated receptor-α (PPAR-α) activator, used since the 1980s as a treatment to reduce triglycerides, improved memory and lowered Aβ levels in murine models of AD [36,37], and has been tested in a phase 1 clinical trial in humans (NCT02045056). Many other cellular pathways associated with neurodegenerative diseases may be targeted [38,39]. However, demonstrating efficacy in these chronic diseases can require decades-long clinical trials, an often frustrating endeavor. Recent GWAS results have even suggested new targets for sporadic prion diseases, such as Jakob disease (formerly called CJD), suggesting even this rapidly progressing, fatal disease may eventually have some therapies [40].

## 7. Testing for Autoimmune Encephalitis

Before planning treatment, it is important to determine any direct relationship between the tumor and dementia symptoms. This is particularly important as certain cancers can cause autoimmune reactions that result in dementia. One of the first of these syndromes to be widely demonstrated resulted from finding autoantibodies to the NMDA receptor in patients with ovarian teratomas who exhibited neurocognitive problems and dementias [41]. Subsequently, many more antigens were discovered that could result in paraneoplastic neurological syndromes, where the antibodies pass the BBB and mistakenly attack normal tissues [42,43]. Autoantibodies in cerebrospinal fluid (CSF) to the AMPA receptor, for example, correlate by up to 60% with cancers, especially of the thymus; those to the GABAa or b receptors have up to a 70% correlation with underlying cancers [44]. Delta/notch-like epidermal growth factor-related receptor (Anti-Tr/DNER) antibodies are found in Hodgkin’s lymphoma; their presence may have a diagnostic role (although the cancer is usually detected first) and may even indicate a relapse [45].

Once autoantibodies are detected, treatment can be started to alleviate the dementia symptoms [43]. This can include removal of circulating antibodies by immunosuppression, high dose steroids, intravenous immunoglobulins (IVIg), and by plasma exchange. Immunosuppressive drugs, azathioprine or mycophenolate mofetil, used in other autoimmune diseases such as lupus, can starve antibody producing cells of purines, or specifically guanine nucleotides. Immunosuppression can also be induced with cyclophosphamide or rituximab. A recent clinical trial (NCT00987389) for the treatment of autoimmune (ANCA)-vasculitis (average age of participants, 63 years) indicated that both plasma exchange and steroid dosing should be minimized [46,47]. Of course, the best treatment is to remove the tumor or eradicate the cancer.

Doctors approach both diagnostics and selection of potential treatments through glasses tinted with experience; they are limited by standard treatment recommendations of the medical profession, and in many cases the insurance industry. Even after the finding of autoantibodies to a given antigen, patients may be prescribed high dose steroids (e.g., 1 g/diem intravenous prednisone, for several days) before any newer, more specific (and usually much more expensive) treatments can be given. Doctors have seen what inexpensive steroid drugs, hailed as wonders when introduced in the late 1940s, can do for chronic autoimmune diseases: psoriasis, multiple sclerosis, and lupus. They are also aware of the problems associated with chronic steroid use, as well as adverse effects of newer treatments. There are now many suggestions for how to evaluate the risks of specific genetically guided treatments [48].

## 8. Genetics May Hold the Key to More Curative Therapy More Suited to Those with Cognitive Impairment

A full genetic profile should be obtained, to guide therapy, while keeping in mind the special needs of the patient. Genetic analysis, to identify specific mutations or clustered ones [49,50], can suggest target treatments for many cancers and chronic diseases of aging. Oncologists are particularly interested in knowing microsatellite status (stable or unstable, MSS or MSI), as well as molecular characteristics of RAS, BRAF, HER2, EGFR, VEGF, MEK, and different kinases that may play a role in tumor progression. MSI (similar to dMMR, see below) tumors can be expected to respond to immunotherapies (IT) as discussed below. The great majority (80–85%) of tumors are MSS, or “cold”. MSS tumors typically do not respond well to IT and other treatments must be sought.

Genetic typing of cancers can hold the clue to their treatment, with oral therapies and infusion or immunotherapies (IT) more suitable for frail individuals with cognitive difficulties, assuming the involvement of helpers who can ensure compliance. Less invasive does not mean less effective; some oral therapies may be superior to intravenous ones [51]. While a recent study of breast cancer patients indicated that those with severe cognitive difficulties were much less likely to have surgery, their breast cancer related survival rate was not significantly affected when appropriately matched to cognitively normal women [52].

## 9. The Growing Promise of Immunotherapies (IT) in Cancer

The development and introduction of new medicines can move at lightning speed or take years of clinical trials, and therapies that look promising can fall by the wayside or become standards of care. In 1999, only 25% of metastatic (stage IV) melanoma patients survived, despite treatment with up to four different drugs [53]. By 2020, new IT drugs, such as a combination of CTLA-4 and PD-1 inhibitors (ipilimumab and nivolumab), have increased the one year survival to above 80%, with four year rates > 50% [54].

Complete genome sequencing of cancer patients should one day be routine. New technologies to keep up with all the data generated include machine learning and AI [50]. There is now a whole toolbox of computer tools for calculating mutations (SIGPRofiler).

Immunotherapies, given as infusions over a course of months, have greatly changed the cancer field, in many cases replacing surgery as the first line of treatment. Although the first immunotherapies, with anti-PL1 or Car-T, were also accompanied by severe side effects, their use in patients who had failed many other therapies was shown to give “miraculous” cures. Today, tumor sequencing and the revelation of targetable neoantigens have clarified some of the reasons for these early mixed results [55].

There was a startling revelation with respect to patients in which Keytruda and related immunotherapies may be most likely to work, differentiating them from those where alternative therapies would be needed. One reason for treatment failure may be that the mutation in the oncoprotein that is driving the cancer may not be recognized by activated natural killer T-cells. This means that small mutations, such as the point mutations G12C or D [56], that can activate K-Ras and other oncogenes, may not change the structure of the protein enough to be recognized. Thus, tumors caused by mutations involving frameshifts [56], where the overall structure of the oncogene dictated protein is significantly changed (enough to generate “neoantigens”), are more likely to be recognized and eliminated. This means that the sickest patients, and those who have failed all the previous treatments, may be the best candidates for immunotherapy.

For example, patients with early-stage colon cancer with mutations in DNA mismatch repair (dMMR) [57] responded very well to IT. In the NICHE-2 trial (a four week course, one treatment with ipilimumab (Yervoy) and two with nivolumab (Opdivo)), 95% of patients (median age of 60 years) with dMMR advanced colon cancer had a major pathologic response (MPR), and 67% had a pathologic complete response (pCR), to immunotherapy without surgery. Other PD-1 inhibitors have also been shown to be very effective in dMMR CRC [58]. A 100% response has been reported for treatment with another IT agent, dostarlimab [59].

However, only about 10–15% of CRCs are dMMR [60] or MSI. The high activity of IT in a discrete group of patients may explain the confusing results of early trials, where IT was used without genetic guidance. In other CRC patients, whose tumors were MMR competent (pMMR), only 4/15 (27%; 95% exact CI: 8–55%) showed pathological responses to IT, with three MPRs and one partial response [57]. Thus, surgery and other therapies may be required for most CRC patients.

Complicating the picture further, it should be noted that immunosuppressive treatments (e.g., for paraneoplastic neurological syndromes) have opposing effects to IT, and that immune stimulating treatments can also cause autoimmune reactions. It is difficult to predict exactly how the immune system will react to future challenges. For example, vaccination after successful IT treatment with nivolumab can cause an immune system overreaction [61]. After being cured of his melanoma with nivolumab, a patient subsequently received two doses of Covid RNA vaccine. Due to an immune system overreaction, he developed Type one diabetes (his blood glucose went to 655 mg/dl and his H1C to > 8) with almost no insulin secretion (as determined by the level of C reactive protein in his urine and sera) [62]. Similarly, adverse reactions to IL-2, an adjunct in some IT trials [63] (or to treatment with its inhibitor daclizumab [64]), can have fatal consequences.

## 10. Therapy Targeted at Other Driver Mutations

While long infusions may be difficult to explain and administer to those with cognitive and movement disorders, targeted therapies may offer easier oral treatments. There are now many different potential therapies for cancer treatment, targeting different tumor antigens. For acute myeloid leukemia (AML), the patient’s genetic profile should be obtained first, even if this means delaying the start of the treatment, as recommendations are different if the disease has unfavorable markers (e.g., mutations in the tumor suppressor TP53), is secondary to a previous treatment, and if the patient is “fit” or “unfit” [65]. Eprenetapopt, a small molecule that may stabilize the TP53 mutated protein to restore its function, in combination with azacytidine, is now in testing for AML [66]. In esophageal cancer, a better marker for response to eprenetapopt (also in combination with chemotherapy) may be expression of SLC7A11 (solute carrier family 7, member 11), specific for cysteine/glutamate transport [67].

Genetic profiling can allow treatment with reduced side effects, by replacing or lowering the dosage of chemotherapy by drugs targeting mutated *EGFR*, *ALK*, *RET*, *ROS1*, *NTRK*, and *MET.* Chemotherapy can also be better targeted by mutation profiling. Patients with *EGFR* mutations have double the rate of response to carboplatin and paclitaxel than those without, while chemotherapy combined with first-generation EGFR tyrosine kinase inhibitors may also be more effective. Treating melanoma with mutations in genes for two kinases on the same pathway, BRAF and MEK, with specific inhibitors, can be used in place of chemotherapy. After years of reports that KRAS was undruggable, there are now therapies in clinical trials for tumors driven by mutations such as KRAS^G12C^ (glycine 12 to cysteine) [68]. Interferons (IFN) [69] may still be prescribed for certain cancers, such as tyrosine kinase resistant chronic myeloid leukemia (CML), with warnings of the possibility they can cause depression after chronic administration. Sometimes these older drugs may be combined with a newer one to achieve a better therapy, as when IFNs [70] may be combined with omacetaxine [71], or ropeginterferon α2b with imatinib [72]. Meanwhile, the usefulness of the cerebron expression measurement in elderly patients taking lenalidomide, a thalidomide derivative [73], as part of a combination therapy [74] is somewhat controversial [75], as this drug can affect many pathways.

As newer, specific therapies are more expensive, they may not be used as frequently in those who face an uncertain future due to what may be perceived as a terminal dementia. A doctor must balance the likelihood of cure and associated side effects against their cost, and whether they are appropriate for the age, complicating illnesses, and overall condition of the patient.

## 11. Conclusions

Individual assessment, open discussion with caregivers and family, and genetic analysis should be used to distinguish cognitive or behavioral difficulties caused by progressive, incurable disease processes, from those that may respond to immuno- and cancer therapies. Dementia should not exclude the patient from cancer treatment, although oral and immunotherapies may be chosen over major surgical treatment in frail individuals. Tumor related immunoresponses should be considered as a cause of cognitive difficulties, especially in those with rapid symptom onset. Treatment costs are also important, as they may be affected by financial considerations based on anticipated years of remaining life. All features of a patient’s life will affect the ideal treatment plan for individuals diagnosed with cancer who are experiencing cognitive problems.

## Figures and Tables

**Figure 1 cancers-15-01055-f001:**
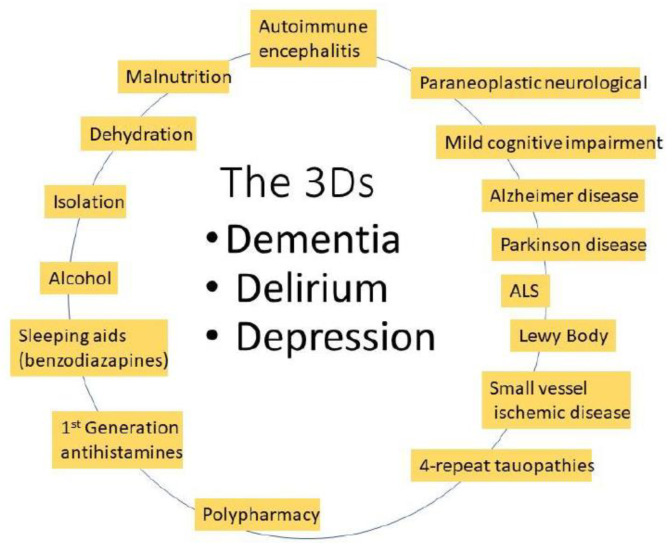
Many different environmentally related or progressive neurological syndromes can have similar “3D” symptoms.

**Table 1 cancers-15-01055-t001:** More than half of cancers are diagnosed in people older than 60 years. The table lists cancers whose median age of diagnosis is above 60 years. Except for ovarian (54–65), most cases are diagnosed between the ages of 65–74.

Cancer	Average Age of Diagnosis
Acute myeloid leukemia (AML)	68
Chronic Lymphocytic Leukemia (CLL)	70
Bladder	73
Breast (BC)	62
Colo-Rectal (CRC)	66
Lung	71
Melanoma	65
Ovarian	63
Pancreatic	70
Prostate	67
Ovarian	63

**Table 2 cancers-15-01055-t002:** Syndromes of aging that may also accompany AD and PD [4].

Abbreviation	Syndrome	Neurological Diagnosis Based on:
PART	primary age related tauopathy	Tau tangles without clear Aβ plaques
AGD	argyrophilic granular disease	Silver staining needles containing 4-repeat tau isoforms (exon 10 immunostaining)
FTLD	frontotemporal lobar degeneration	4-repeat tauopathy
PSP	progressive supranuclear palsy	4-repeat tauopathy; Midbrain atrophy (“hummingbird sign” on MRI)
HSA	Hippocampal Sclerosis of Aging	Shrinkage of the hippocampus (seen in 30% of those above 85 even without symptoms)
SVID	small vessel ischemic disease	Vascular inclusions

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
