# Peer review of "Distinguishing Curable from Progressive Dementias for Defining Cancer Care Options"

_cancers, 2023, doi:10.3390/cancers15041055_

Round 1
Reviewer 1 Report (Previous Reviewer 1)
After looking over the authors' response to my comments, I feel satisfied and have no further comments, which leads to the conclusion that the paper is qualified for acceptance to your Journal.
Reviewer 2 Report (Previous Reviewer 2)
None to add
This manuscript is a resubmission of an earlier submission. The following is a list of the peer review reports and author responses from that submission.
Round 1
Reviewer 1 Report
Thank you for the opportunity to review the paper. This paper focused more on genetics and immune therapy among demeted individuals with cancer. The followings are my comments for the paper.
Overall, despite the chance of receiving such therapies, discussions on shared decision making is required in the clinical settings. If the patient cannot offer their thoughts whether or not to take treaments less invasive, families and other members in the medical care team should share their opinion alternatively. Please give thoughts about this topic. When making discussion about this topic, scientific evidence on the prognosis of demented patients taking cancer treatment are also required.
P1 L16
What does “deprescribing” mean here? Terms in the abstract should be clear by just reading the abstract.
P4 Paragraph 1
Will it still be appropriate for the older patient to receive brain imaging although it does not lead to prescription? I would like to hear the authors’ opinion about this topic.
Reviewer 2 Report
The topic is of importance, but not faced in appropriate way from a geriatric point of view in my opinion. In particular, the most important topic, i.e., the importance of dementia/MCI is clinical decision making in older people having cancer was not discussed.
Minor comments:
-Please don't use demented, it could be considered offensive
-deliria? you mean delirium